# RNA-Seq Analysis Revealed the Virulence Regulatory Network Mediated by the Ferric Uptake Regulator (Fur) in *Apostichopus japonicus* Pathogenesis Induced by *Vibrio splendidus*

**DOI:** 10.3390/microorganisms13061173

**Published:** 2025-05-22

**Authors:** Changyu Liao, Lincheng Hu, Si Zhu, Weikang Liang, Lei Yang, Chenghua Li

**Affiliations:** 1State Key Laboratory of Agricultural Products Safety, Ningbo University, Ningbo 315211, China; liaochangyu@nbu.edu.cn (C.L.); hlc282503@163.com (L.H.); zhusi@nbu.edu.cn (S.Z.); liangweikang@nbu.edu.cn (W.L.); 2Laboratory for Marine Fisheries Science and Food Production Processes, Qingdao Marine Science and Technology Center, Qingdao 266237, China

**Keywords:** ferric uptake regulator, virulence, transcriptome sequencing, flagellar assembly, skin ulcer syndrome

## Abstract

The uptake and utilization of iron by bacteria must be strictly controlled. The ferric uptake regulator (Fur) is a global transcription factor widely present in bacteria that can perceive cellular iron levels and adjust the expressions of various genes accordingly. Our earlier research demonstrated that the knockdown of the *fur* gene in *Vibrio splendidus* significantly reduced its lethality to *Apostichopus japonicus*. Although the functions and mechanisms of Fur in regulating bacterial virulence genes have been extensively studied, its virulence regulatory network during *V. splendidus* pathogenesis in *A. japonicus* remains unclear. In this article, transcriptome sequencing analysis of *V. splendidus* under different iron conditions reveals substantial differential gene expressions in the simulated pathogenic environments, identifying 1185 differentially expressed genes, including 198 downregulated and 987 upregulated genes. Comparative analysis between wild-type and *Vsfur* knockdown strains shows that *Vsfur* knockdown altered the expression of 3593 genes in *V. splendidus*, with the most significant differential expression observed under simulated pathogenic conditions (1030 upregulated and 72 downregulated). KEGG enrichment analysis indicates that *Vsfur* knockdown caused significant gene enrichment in the flagellar assembly pathway and bacterial secretion system, critically impairing flagellar synthesis and secretion system function in *V. splendidus*. Eight genes selected for qRT-PCR validation showed expression levels in line with the RNA-seq results. Consistent with the transcriptomic results, *Vsfur* knockdown resulted in reduced antioxidant capacity, bacterial competitiveness, and cytotoxicity in *V. splendidus*. These findings elucidate the virulence regulatory mechanism of Fur in *V. splendidus* and provide a reference for understanding the occurrence of *A. japonicus* skin ulcer syndrome.

## 1. Introduction

Sea cucumbers, belonging to the class Holothuroidea within the phylum Echinodermata, are marine invertebrates primarily distributed in the western Pacific Ocean, including the Yellow Sea, Sea of Japan, and Sea of Okhotsk. With the expansion of aquaculture scale and density, diseases have become increasingly prominent, gradually emerging as an obstacle to the healthy development of the sea cucumber industry [1]. These diseases have caused significant economic losses to aquaculture farmers and posed serious threats to the ecological environment and food safety [2]. Therefore, developing green and healthy disease prevention and control strategies for sea cucumbers is of great importance. Sea cucumber diseases are characterized by their cyclical nature and polymorphism, with outbreaks caused by protozoans, fungi, bacteria, and viruses. Currently, bacterial diseases are the most frequently reported, primarily including skin ulcer syndrome (SUS), gastric rot disease, edge necrosis, and evisceration syndrome [3]. Among these, SUS is the most prevalent and devastating disease in the sea cucumber industry, attracting widespread attention from researchers. Pathogens currently identified as causing SUS in sea cucumbers include *Vibrio* spp., *Pseudoalteromonas* spp., *Aeromonas hydrophila*, etc. [4,5].

*Vibrio splendidus* is a Gram-negative bacterium that widely exists in the marine environment and has been identified as the primary pathogen responsible for SUS in sea cucumbers (*Apostichopus japonicus*). In recent years, significant advances have been made in the study of *V. splendidus* virulence factors. Liang et al. demonstrated that the virulence factor Vshppd plays a dual role in pathogenicity and stress resistance during *V. splendidus* invasion of *A. japonicus* [6,7]. Transcriptomic sequencing further elucidated the pathogenic pathway mediated by Vshppd, revealing that Vshppd may regulate virulence by influencing flagellar gene expression through modulating energy metabolism via the tricarboxylic acid (TCA) cycle [7]. Research on host–pathogen interactions between *V. splendidus* and *A. japonicus* have yielded significant insights. Dai et al. uncovered the molecular mechanism by which the eukaryotic-like factor STPKLRR of *V. splendidus* targets the phosphorylated cytoskeletal protein AjTmod in *A. japonicus*, inducing the dissociation of the AjTmod/Actin complex to promote bacterial internalization [8]. Additionally, another eukaryotic-like factor, Sppsk1, facilitates *V. splendidus* phagosome escape by suppressing phagosome acidification and maturation through its kinase domain and phosphatase domain, respectively [9]. These findings provide a detailed explanation of the mechanisms underlying *V. splendidus* internalization and immune evasion during infection. In terms of bacterial virulence regulation, short-chain acyl-homoserine lactone signaling molecules and indoles have been detected in *V. splendidus*, both of which modulate virulence gene expression [10]. Furthermore, the ferric uptake regulator (Fur), a global regulatory protein, coordinates gene expression to optimize bacterial survival and virulence under fluctuating iron conditions, thereby playing a critical role in environmental iron adaptation [11].

Notably, Fur plays a pivotal role in controlling virulence-associated gene expression during host invasion with many pathogens, including processes such as oxidative stress response, quorum sensing, swarming motility, and biofilm formation [12,13]. In *Vibrio vulnificus*, Fur influences the entire life cycle of the pathogen, from its survival in marine environments to its persistence in host blood [11]. Our previous study demonstrated that the knockdown of the *fur* gene in *V. splendidus* significantly reduced its lethality to *A. japonicus* [14]. However, the virulence regulatory network governed by Fur in *V. splendidus* during *A. japonicus* infection remains uncharacterized. In the present study, transcriptome sequencing was performed on *V. splendidus* samples treated under high-, low-, and normal-iron, and simulated infection coelomic fluid conditions from *A. japonicus* to investigate gene expression changes under varying iron concentrations. Furthermore, transcriptome sequencing of the wild-type and *Vsfur* mutant strains under high-, low-, and normal-iron, and coelomic fluid-treated environments was conducted to elucidate the key pathogenic pathways regulated by Fur in *V. splendidus*. Our study aims to characterize the key virulence genes regulated by Fur during the *A. japonicus* infection process, thereby providing a theoretical foundation for SUS prevention and control.

## 2. Materials and Methods

### 2.1. Animals, Bacterial Strains, and Culture Conditions

Sea cucumbers were purchased from Ningbo Bowang Aquaculture Co., Ltd. (Ningbo, China), with a specification of 130 ± 10 g. They were acclimatized in PVC tanks for one week under controlled salinity (28 ± 2‰) and water temperature (16 ± 1 °C), with continuous aeration provided using an oxygen pump. Coelomic fluid extraction was performed as follows: the *A. japonicus* body surface was disinfected with medical alcohol, followed by dissection using sterilized scissors and forceps. Subsequently, the coelomic fluid was filtered through a 200-mesh nylon mesh. The *V. splendidus* strain was dissociated from lesion tissues of sea cucumbers naturally infected with SUS, and its pathogenicity was confirmed through reinfection experiments [15]. This *V. splendidus* strain AJ01 was deposited in the China General Microbiological Culture Collection Center (CGMCC) under accession number CGMCC No. 7.242. The *Vsfur* knockdown strain (Δ*Vsfur*) was constructed in our laboratory [14]. *V. splendidus* was cultured at 28 °C in the 2216E medium. To prepare the 2216E medium, 5.0 g tryptone and 1.0 g yeast extract were mixed and adjusted to 1000 mL with seawater (pH 7.6–8.0). To prepare the solid medium, 3.0 g of agar powder was added to 250 mL of the solution, which was sterilized via autoclaving. For the iron-supplemented PBS buffer (50 μM FeCl_3_), 1.622 g FeCl_3_ powder was dissolved in 10 mL sterile water to prepare a 1.0 mM stock solution. Five hundred microliters of this stock was added to the 10 mL PBS buffer. For the iron-depleted PBS buffer (100 μM 2,2′-bipyridine), 1.562 g 2,2′-bipyridine powder was dissolved in 10 mL of sterile water to prepare a 1.0 mM stock solution. One thousand microliters of this stock was added to 10 mL of PBS buffer.

### 2.2. Sample Collection

The *V. splendidus* strain was inoculated onto a 2216E solid agar plate. After 24 h of incubation, a single colony was selected from the plate and inoculated into a conical flask containing the fresh 2216E medium. The culture was incubated at 28 °C with shaking at 180 rpm. The optical density (OD) of the bacterial suspension was measured at regular intervals using a UV spectrophotometer. When the OD_600_ reached 1.0, equal volumes of the bacterial culture were aliquoted into 2 mL RNase/DNase-free centrifuge tubes. The bacterial cells were collected by centrifugation at 8000× *g* for 5 min at room temperature (RT). The pellets were resuspended as follows: high iron concentration group (H group)—1 mL of PBS buffer with a high iron concentration; low iron concentration group (L group)—1 mL of PBS buffer with a low iron concentration; coelomic fluid treatment group (C group)—900 μL of PBS buffer supplemented with 100 μL of *A. japonicus* coelomic fluid; normal iron concentration group (N group)—1 mL of PBS buffer with a normal iron concentration (control group). The experimental groups included the Δ*Vsfur* mutant strain (group 1) and the wild-type AJ01 strain (group 2). The resuspended bacterial cells were further incubated at 28 °C with shaking for 1 h. After centrifugation at 8000× *g* for 5 min to remove the supernatant, the bacterial pellets were flash-frozen in liquid nitrogen for 3–4 h. The samples were then transferred to a −80 °C freezer for storage. During transportation, the samples were embedded in sufficient dry ice and subsequently subjected to prokaryotic reference transcriptome sequencing analysis at Sangon Biotech (Shanghai) Co., Ltd. (Shanghai, China).

### 2.3. RNA Extraction and Quality Control

RNA was extracted using the Total RNA Extractor (Trizol) Kit (Takara, Beijing, China) according to the manufacturer’s protocol. Briefly, the lysed samples were incubated at RT for 5–10 min to dissociate nucleoprotein complexes. Subsequently, 0.2 mL of chloroform was added, followed by vigorous vortexing for 15 s and incubation at RT for 3 min. After phase separation by centrifugation (12,000× *g*, 4 °C, 10 min), the aqueous phase was transferred to a fresh centrifuge tube and mixed with an equal volume of isopropanol. The mixture was incubated at RT for 20 min to precipitate RNA. After centrifugation at 12,000× *g*, under 4 °C for 10 min, the supernatant was discarded. The RNA pellet was washed with 1 mL of 75% ethanol (12,000× *g*, 4 °C for 3 min). The pellet was air-dried at RT for 5–10 min and dissolved in 30–50 μL of RNase-free ddH_2_O. Finally, the RNA concentration was quantified using a Qubit 2.0 Fluorometer (Shanghai, China). RNA integrity and genomic DNA contamination were assessed via agarose gel electrophoresis.

### 2.4. cDNA Library Construction and Sequencing

Ribosomal RNA (rRNA) depletion was performed via DNA probe-guided RNase H enzymatic digestion, followed by DNase I treatment to eliminate residual DNA. Purified mRNA and non-coding RNA were fragmented under elevated temperature and reverse-transcribed into double-stranded cDNA using random hexamers and reverse transcriptase. The cDNA underwent end repair, adenylation, and adapter ligation for library preparation. Fragment size selection was performed using DNA Clean Beads. PCR amplification with indexed primers enriched adapter-ligated fragments. The quality of the constructed libraries was assessed via electrophoresis on an 8% polyacrylamide gel. Recovered DNA quantification was validated using the Qubit DNA Assay Kit to ensure precise measurements. Libraries were pooled in equimolar ratios (1:1) and subjected to sequencing on the Illumina platform.

### 2.5. Sequence Data Analysis and Functional Annotation

The raw sequencing data were quality assessed using FastQC. Quality trimming was performed with Trimmomatic to obtain high-quality and reliable data. In this study, Bowtie2 software (version 2.3.2) was utilized to analyze the filtered sequencing data, aligning the sequencing data against the *V. splendidus* genome ASM2281204v1, thereby completing sequence annotation. Functional annotation was performed via BLASTp against the NCBI non-redundant (NR), Kyoto Encyclopedia of Genes and Genomes (KEGG), and Gene Ontology (GO) databases. We utilized the topGO software (version 2.24.0) to conduct Gene Ontology (GO) enrichment analysis and employed the clusterProfiler package to analyze the enrichment of KEGG pathways and COG (clusters of orthologous groups) classifications. Additionally, we constructed network diagrams based on the data from these gene function enrichments to illustrate the connections between different biological processes and pathways.

### 2.6. Gene Expression Difference Analysis

Using the bioinformatics tool DESeq2, we conducted an in-depth comparative analysis of gene expression profiles across different samples. We counted the mapped reads from wild-type AJ01 and Δ*Vsfur* mutant strains aligned to assembly contigs in different iron environments and normalized them to FPKM (fragments per kilobase per million mapped reads). Then, we calculated the differential expression abundance of each gene on a contig with a Q value ≤ 0.05 and an absolute value of log_2_ (fold change) ≥ 1. We performed data visualization to intuitively visualize the analytical results. Based on the statistical results of differentially expressed genes (DEGs), Venn diagrams were generated to reveal shared characteristics and unique differences in the gene expression profiles under distinct conditions. Heatmaps were constructed to comprehensively display the hierarchical structure and expression patterns of the genes. Additionally, cluster analysis was performed to further explore potential associations and expression pattern similarities among the genes.

### 2.7. RNA Isolation and qRT-PCR

To validate transcriptome sequencing accuracy, this study selected eight genes with significant differential expression for quantitative real-time PCR (qRT-PCR) analysis. Gene-specific primers were designed using Primer Premier 5.0 software based on the known sequences, with detailed information provided in Table 1. In the qRT-PCR assay, SYBR Green fluorescent dye was used to quantify gene expression in *V*. *splendidus* under different iron concentrations. Primers 933F and 16SRTR1 targeting 16S rDNA were employed as the reference gene to normalize experimental variations. The amplification procedure was as follows: 95 °C for 1 min, followed by 40 cycles of 95 °C for 5 s and 60 °C for 34 s. The PCR reaction mixture was prepared as follows: Dye-II (ROX, 0.4 μL), qPCR forward primer (10 μM, 0.8 μL), qPCR reverse primer (10 μM, 0.8 μL), TB Gree^TM^ Premix Ex Taq II (10 μL), and cDNA template (8 μL), with a total reaction volume of 20 μL. The reaction mixture was promptly prepared under light-protected conditions. Each sample was briefly centrifuged to ensure thorough mixing of the components. Subsequently, five biological replicate samples from each experimental group were loaded into eight-strip tubes and analyzed using an ABI 7500 Quantitative Real-time PCR System (Thermo Fisher Scientific, Waltham, MA, USA). The 2^−ΔΔCt^ method was applied to accurately evaluate the experimental results for data analysis.

### 2.8. H_2_O_2_ Stress, Bacterial Interspecies Competition, and Cytotoxicity Assay

The H_2_O_2_ stress assay was performed as follows: single colonies of the activated wild-type AJ01 and Δ*Vsfur* mutant strains were picked from solid plates and inoculated into 5 mL of the fresh 2216E liquid medium. Cultures were grown until they reached an OD_600_ of 0.6–0.8. For each strain, 1 mL of a culture was aliquoted into two sterile 1.5 mL EP tubes labeled as the NC (negative control) and treatment groups. The treatment group was supplemented with a H_2_O_2_ solution at a final concentration of 10 μM. Both groups were incubated in the dark at 220 rpm and 28 °C for 10 min. The H_2_O_2_-treated bacterial suspensions were then serially diluted in sterile PBS. Aliquots of each dilution were spotted onto labeled 2216E solid plates (without antibiotics). Following overnight incubation, bacterial colonies were enumerated. For the bacterial interspecies competition assay, the pET-28a plasmid was transformed into *Escherichia coli* BL21 cells to induce kanamycin resistance. Activated cultures of *E. coli* BL21, wild-type AJ01, and Δ*Vsfur* strains were adjusted to 1 × 10^7^ CFU/mL and resuspended in sterile PBS. Equal volumes of the three bacterial strains were co-cultured in LB liquid medium for 3 h. Viable bacterial counts were determined using the serial dilution method. For the cytotoxicity assay, the CCK-8 kit (Beyotime Biotechnology, Shanghai, China) was used to assess cytotoxicity. *A. japonicus* coelomocytes were treated with 10 μL of wild-type AJ01 or Δ*Vsfur* mutant bacterial cultures and incubated at 16 °C for 1 h. Subsequently, 10 μL of the CCK-8 solution was added to each well, followed by a 3 h incubation. Absorbance was measured at 450 nm using a microplate reader.

### 2.9. Statistical Analysis

GraphPad Prism software (version 5.00) was used for statistical analyses. The standard deviation was a measure of the variability of several independent variables around the mean. For each of the two groups, an unpaired two-tailed Student’s *t*-test was performed. Two-way analysis of variance (ANOVA) was used for experiments with more than one group and independent variable, while one-way ANOVA was used for analyses with more than one group and a single independent variable. All data are expressed as the mean ± SD and represent the results of at least three independent experiments. Statistical significance was defined as * *p* < 0.05 and ** *p* < 0.01.

## 3. Results

### 3.1. Library Sequencing and De Novo Transcriptome Assembly

Using the Illumina HiSeq 4000 platform, a total of 230,667,284 raw reads were generated. After filtering low-quality reads, 222,138,268 clean reads were retained for subsequent de novo assembly. Detailed quality assessments of the sequencing output for all samples are presented in Table 2. The raw data were submitted to the NCBI SRA database with accession number SUB15272511. For prokaryotic organisms with high gene density, genome alignment analysis of quality-controlled sequencing data was performed using Bowtie2 software. Statistical analyses were further conducted with RSeQC to comprehensively evaluate the alignment results. The experimental results demonstrate that, with the exception of groups C1 and C2, all other groups exhibited a total mapped read rate exceeding 74%.

### 3.2. Identification of Differentially Expressed Transcripts in Different Iron Environments

Transcriptomic analysis revealed significant differences in mRNA abundance between the experimental groups (H, L, and C) and the control group (N). In the treated wild-type samples, the H2 group exhibited 365 differentially expressed mRNAs (205 upregulated and 160 downregulated), the L2 group displayed 303 mRNAs (89 upregulated and 214 downregulated), and the C2 group showed 1185 mRNAs (987 upregulated and 198 downregulated). The C2 vs. N2 comparison demonstrated the most pronounced transcriptional divergence, both in terms of absolute DEG counts and upregulated gene numbers (Figure 1A). Consistent with this, Venn diagram analysis identified 819 unique DEGs in the C2 vs. N2 comparison, which was significantly higher than those in the H2 and L2 groups (Figure 1B). Hierarchical clustering heatmap analysis (Figure 1C) further confirmed distinct expression profiles between the C2 and other groups (H2, L2, and N2). These findings align with the simulated infection process of *V. splendidus* in *A. japonicus*, where pathogenicity involves host cell disruption and nutrient competition. The marked upregulation of genes in the C2 group strongly suggests their functional association with *V. splendidus* virulence mechanisms, potentially driving bacterial adaptation and pathogenicity during host invasion.

### 3.3. Identification of Differentially Expressed Transcripts Regulated by Vsfur

Transcriptomic analysis identified substantial DEGs across the experimental groups. The C2 vs. C1 comparison exhibited the most pronounced transcriptional divergence, with 1102 DEGs (1030 upregulated and 72 downregulated). Other group comparisons showed distinct expression patterns. The H2 group exhibited 730 DEGs relative to H1, comprising 391 upregulated and 339 downregulated genes. Transcriptional profiling further identified 846 DEGs between the L2 and L1 groups, with 437 upregulated and 409 downregulated entries. Moreover, comparative analysis revealed 915 DEGs in the N2 group compared to N1, including 516 upregulated and 399 downregulated transcripts (Figure 2). These findings delineate the iron-dependent regulatory landscape of Fur-mediated DEGs. Volcano plots were used to visualize these differential expression profiles, while Venn diagram analysis delineated shared and unique DEG subsets among the groups (Figure 3A). Hierarchical clustering further distinguished the transcriptional profile of the sea cucumber coelomic fluid-treated group from the others, with notable expression divergence between the wild-type and mutant strains. These findings highlight the critical role of *Vsfur* in modulating adaptive responses to host–pathogen interactions (Figure 3B).

### 3.4. GO and KEGG Enrichment Analysis of Differentially Expressed Genes

Wild-type strains and mutants exhibited similar GO enrichment patterns across biological processes, cellular components, and molecular functions under varying iron conditions. DEGs were predominantly associated with fundamental biological processes, including cellular and metabolic processes. Molecular functions primarily involved catalytic and structural molecule activities (Figure 4). KEGG pathway analysis revealed significant DEG enrichment in ribosomal-associated pathways across all comparison groups. Notably, under the high-iron treatment, DEGs between the wild-type and mutant strains showed preferential enrichment in the glycolysis/gluconeogenesis and TCA cycle pathways. Under iron-deficient conditions, comparative analysis demonstrated relative DEG enrichment in the TCA cycle and alanine, aspartate, and glutamate metabolism pathways. Under simulated pathogenic conditions, the wild-type strains exhibited specific enrichment in the phenylalanine, tyrosine, and tryptophan biosynthesis pathways, along with the lysine biosynthesis pathways compared to the mutants. Under normal iron conditions, DEGs were primarily enriched in the glycine, serine, and threonine metabolism pathways, complemented by TCA cycle activation (Figure 5). Figure 6 displays the bar chart of COG classification for the DEGs, indicating that these DEGs are classified into functional COG categories.

### 3.5. Validation of Differentially Expressed Genes by qRT-PCR

To validate DEGs identified from transcriptomic analysis, eight candidate genes were selected for qRT-PCR verification, including flavodoxin (FldA), preprotein translocase subunit (SecA), tRNA dihydrouridine synthase (DusB), aspartate ammonia-lyase (AspA), carbon storage regulator (CsrA), adenylosuccinate lyase (PurB), flagellar basal body rod protein (FlgC), and carbamoyl-phosphate synthase large subunit (CarB). Melting curve analysis of all PCR products confirmed amplification specificity. A significant concordance was observed between the qRT-PCR results and the transcriptomic data, with all genes exhibiting consistent expression patterns (Figure 7A). These results validate the reliability and accuracy of transcriptomic sequencing and assembly.

### 3.6. Vsfur Gene Deletion Affects Antioxidant Capacity, Bacterial Competitiveness, and Cytotoxicity

To investigate the role of Vsfur in oxidative stress response, the wild-type AJ01 and Δ*Vsfur* strains were exposed to hydrogen peroxide and analyzed via dilution plating. The Δ*Vsfur* mutant exhibited significantly reduced survival rates compared to the wild-type mutant, indicating impaired antioxidant capacity (Figure 7B). To assess interbacterial competitiveness, co-culture assays were performed by mixing 1 mL of *E. coli* BL21 with equal volumes of wild-type AJ01 or Δ*Vsfur* strains in the 2216E medium at 28 °C for 3 h. Viable cell counts quantified through serial dilution demonstrated a marked decrease in the competitive fitness of the Δ*Vsfur* strain (Figure 7C). Acute cytotoxicity was evaluated using a CCK-8 assay. After 3 h of co-incubation with host cells, OD_450_ measurements revealed an inverse correlation between cytotoxicity and absorbance. The Δ*Vsfur* mutant showed significantly attenuated cytotoxicity relative to the wild-type mutant, suggesting that Vsfur is critical for virulence regulation (Figure 7D).

## 4. Discussion

Similarly to other invertebrates, *A. japonicus* lacks adaptive immunity, with coelomocytes playing a central role in its immunological defense [16,17]. The immune response in *A. japonicus* is mediated by coelomocytes and diverse humoral immune factors within the coelomic fluid. *A. japonicus* possesses an expansive coelomic cavity filled with coelomic fluid, which functions analogously to lymph and contains a heterogeneous population of immunocompetent cells. These coelomocytes exhibit functional parallels to vertebrate erythrocytes [18]. Previous studies have predominantly focused on the immune responses of invertebrate coelomocyte populations following pathogenic infection. In this study, we simulated the infection process of *V. splendidus* in *A. japonicus* by co-culturing the pathogen with host coelomic fluid, aiming to elucidate how *V. splendidus* communicate with the external environment to achieve host infection. Moreover, our earlier research demonstrated that the knockdown of the *fur* gene in *V*. *splendidus* significantly reduced its lethality to *A*. *japonicus* [14]. Consequently, we conducted a series of experiments to investigate the underlying mechanisms of this phenomenon.

Firstly, we analyzed the DEGs of wild strains under different iron environments. Comparative analysis demonstrated significantly higher numbers of DEGs in high-iron (H group) and simulated infection (C group) conditions compared to iron-limited environments (L group). This observation may reflect growth retardation and metabolic suppression during iron deprivation, whereas iron overload and infection-mimicking conditions necessitate comprehensive genomic adaptations for environmental acclimatization [19,20,21]. GO and KEGG enrichment analyses showed that DEGs were enriched in biological processes, such as metabolism, cellular processes, and stress responses, as well as pathways, including the TCA cycle, oxidative phosphorylation, and pyruvate metabolism. These findings suggest the iron-concentration-dependent modulation of energy metabolism and stress-responsive mechanisms in *V. splendidus*. TCA cycle analysis further identified iron-responsive genes, notably *sdhD*, encoding a hydrophobic subunit of succinate dehydrogenase that forms a dimer with *sdhC* to bind coenzyme Q in the electron transport chain [22]. In *V. splendidus*, the *sdhC/D* operon is iron-regulated, with upregulated expression under simulated pathogenic conditions potentially enhancing respiratory efficiency and antioxidant capacity. Similarly, aceE expression in the experimental group was upregulated compared to the control group. The aceE gene (pyruvate dehydrogenase E1 component) forms an operon with aceF, encoding a hydrophilic matrix-localized catalytic subunit critical for pyruvate decarboxylation to acetyl-CoA. This oxygen-sensitive complex serves as a metabolic nexus between glycolysis and the TCA cycle, with transcriptional regulation influenced by multiple environmental factors [23]. In *Mycoplasma synoviae*, dihydrolipoamide dehydrogenase (*dld*) is not only a biological enzyme, but also an immunogenic surface-exposed protein that can adhere to host cells. Consistent with our previous research, the multifunctional *dld* gene participates in diverse metabolic pathways and bacterial adhesion, directly linking to pathogenicity [24,25]. It is also significantly upregulated after *V. splendidus* stimulation in the coelomic fluid. The upregulation of these genes suggests their potential involvement in either *V. splendidus*’ adaptation to iron environmental changes or pathogenic processes.

Directed acyclic graphs (DAGs) were employed to visualize GO enrichment patterns, effectively illustrating hierarchical relationships and enrichment significance among GO terms [26,27]. Comparative analysis revealed conserved cellular component remodeling in the H and L groups versus the N group, with progressive enrichment from cellular components (GO:0005575) to cytoplasmic ribosomes (GO:0022626) and small ribosomal subunits (GO:0015935), suggesting iron-regulated ribosomal adjustments for protein synthesis [28]. Notably, the L group exhibited significant enrichment of bacterial-type flagellar stator complexes (GO:0120101). Divergent molecular function patterns emerged between the L and C groups. The L group showed multifunctional enrichment spanning metal ion binding (GO:0046872), GTP cyclohydrolase II activity (GO:0003935), and 2-isopropylmalate synthase activity (GO:0003852). In contrast, the C group displayed focused enrichment in ribosomal constituents (GO:0003735), RNA binding (GO:0003723), and nucleotide binding (GO:0000166), aligning with infection-associated pathways, including the TCA cycle, oxidative phosphorylation, and quorum sensing. The following critical virulence-associated genes exhibited marked upregulation: 4-hydroxyphenylpyruvate dioxygenase and type II secretion system protein [7,15]. Strikingly, the C group demonstrated coordinated induction of flagellar biosynthesis operons, with encoded proteins that facilitated host colonization and pathogenicity via motility regulation and adhesion complex formation [29,30,31,32].

Comparative DEG analysis between the wild-type and mutant strains demonstrated that Fur acts as a global transcriptional regulator governing diverse biological processes and molecular functions [33]. GO enrichment analysis revealed consistent enrichment patterns across all iron conditions, underscoring the broad and uniform impact of *Vsfur* knockdown on gene expression. DEGs were predominantly enriched in biological processes, such as cellular processes, metabolism, and cellular component organization, which regulate growth, stress responses, and cell division. At the cellular component level, DEGs were localized to the cell membrane, cell wall, ribosomes, and flagella, indicating changes in structural and morphological. Molecular function analysis highlighted roles in catalytic and structural molecule activity and binding, implicating enzymes, transcription factors, and binding proteins in *Vsfur*-mediated regulation. KEGG analysis further revealed iron-independent enrichment of ribosome-related pathways, as *Vsfur* knockdown dysregulated ribosomal synthesis and activity [34,35]. Subunit-specific analysis showed altered expressions of 50S and 30S ribosomal subunits, including genes encoding *RF1* (termination), *IF3* (initiation), *EF-Tu* (tRNA delivery), *EF-G* (peptide bond formation), and *FtsY/Ffh* (signal recognition) [36,37], indicating that *Vsfur* controls various protein synthesis stages. Concurrently, *Vsfur* knockdown enriched flagellar assembly genes (*FlgC*, *FlgB*, *FlgD*, *FlgK*, *FliE*, and *FliT*), with significant upregulation in wild-type strains versus knockdown mutants, linking Δ*Vsfur* to motility and virulence attenuation [38]. The multifaceted functional impairments observed in the Δ*Vsfur* strain underscore the biological significance of the Fur regulatory network. Phenotypic characterization revealed three core deficiencies: compromised oxidative stress resistance, diminished competitive efficacy against bacterial competitors, and significantly attenuated cytotoxicity. *Pseudomonas aeruginosa* modulates host NADPH oxidase activity to suppress ROS generation and enhance survival [39], while *Vibrio cholerae* employs type VI secretion system effectors to eliminate intestinal microbiota for ecological niche dominance [40]. These pleiotropic phenotypes demonstrate the central role of Fur in coordinating both core metabolic functions and context-dependent virulence adaptations.

## 5. Conclusions

In summary, this study elucidates the pivotal role of Fur in *V. splendidus* virulence regulation. Systematic investigation into the Fur-mediated regulatory network revealed its essential role in both bacterial survival and pathogenicity. Fur mechanistically orchestrates virulence determinants through the transcriptional modulation of quorum sensing systems, two-component signal transduction pathways, bacterial secretion system, and flagellar biosynthesis clusters. Notably, we identified iron availability-dependent mechanistic divergence in Fur-mediated regulation. Our results prove the virulence regulation mechanism of *V. splendidus* and provide an important reference for the development of precise drugs to block *V. splendidus* virulence. In future research, the development of transcription factor modulators or quorum-sensing inhibitors may enable innovative applications in aquaculture disease prevention and control strategies.

## Figures and Tables

**Figure 1 microorganisms-13-01173-f001:**
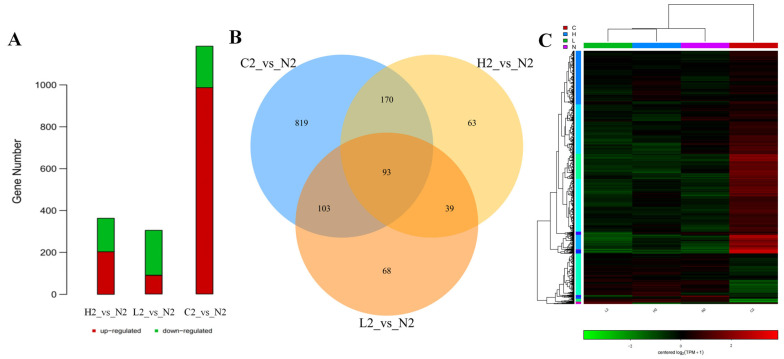
Differential gene expression analysis results. (**A**) The figure is a statistical bar chart that displays the names of various differential comparisons and their distribution in the number of up- and downregulated differential genes. Green and red represent down- and upregulated genes, respectively. (**B**) Differential gene Venn diagram. The numerical markers in the figure reveal the number of specific or shared differentially expressed genes (DEGs). (**C**) Cluster heatmap of differential gene expression levels. The vertical and horizontal axes in the graph represent each gene and different samples, respectively. The depth of color intuitively reflects gene expression strength in a specific sample. Red and green indicate relatively high and low expression levels of the gene in the corresponding sample, respectively. The left side of the graph shows the clustering relationships between genes, presented in the form of a tree diagram.

**Figure 2 microorganisms-13-01173-f002:**
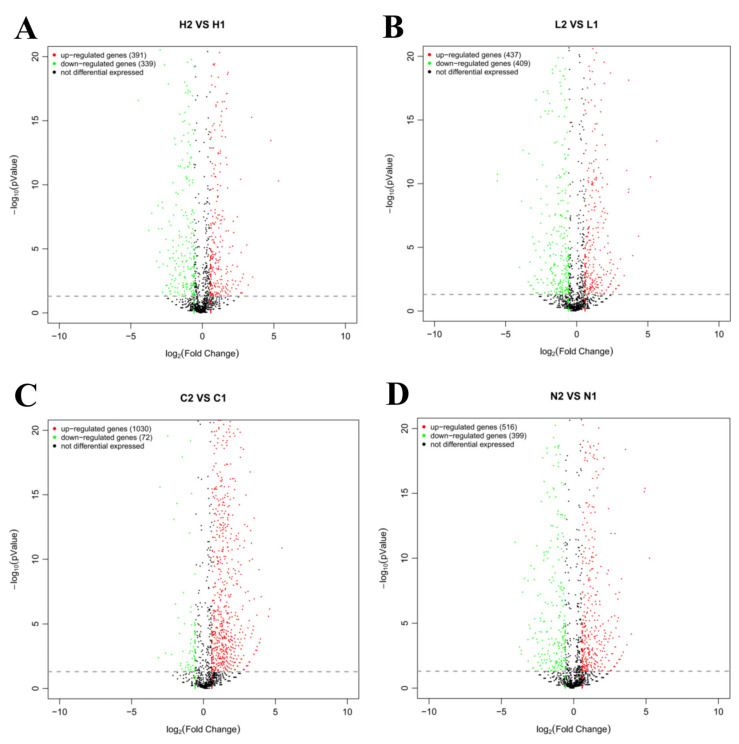
Volcanic maps of the comparison groups. The figure shows the statistical results of gene expression differences between different groups of samples. The horizontal axis represents the multiple changes in gene expression, calculated as the fold change and presented in log (B/A) form. The vertical axis displays the statistical significance of changes in gene expression, i.e., *p* value. The red and green dots represent up- and downregulated genes, respectively. The black dots represent non-DEGs. (**A**) H2 vs. H1; (**B**) L2 vs. L1; (**C**) C2 vs. C1; (**D**) N2 vs. N1.

**Figure 3 microorganisms-13-01173-f003:**
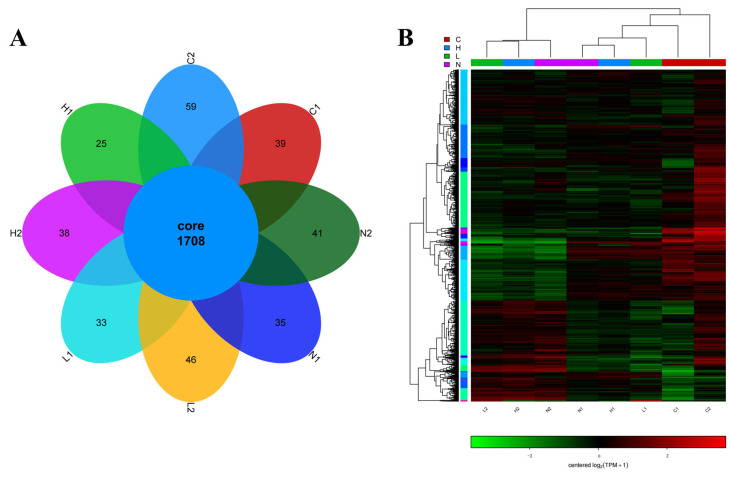
(**A**) Venn diagram of differentially expressed genes. The various parts in the figure are distinguished by different colors to better display different comparison groups. The numbers in the figure are used to represent the specific or shared number of DEGs. Among them, overlapping and non-overlapping regions represent the common and unique number of DEGs among different comparison groups, respectively. (**B**) Each row and column represent a unique gene and a specific sample, respectively. Red and green represent relatively high and low expression levels of genes in the sample, respectively. The tree chart reveals the similarity between genes: when the branches of two genes on the tree chart are close to each other, it indicates that their expression levels are very close. The names of all samples are listed below the figure for easy identification and tracking of information for each sample.

**Figure 4 microorganisms-13-01173-f004:**
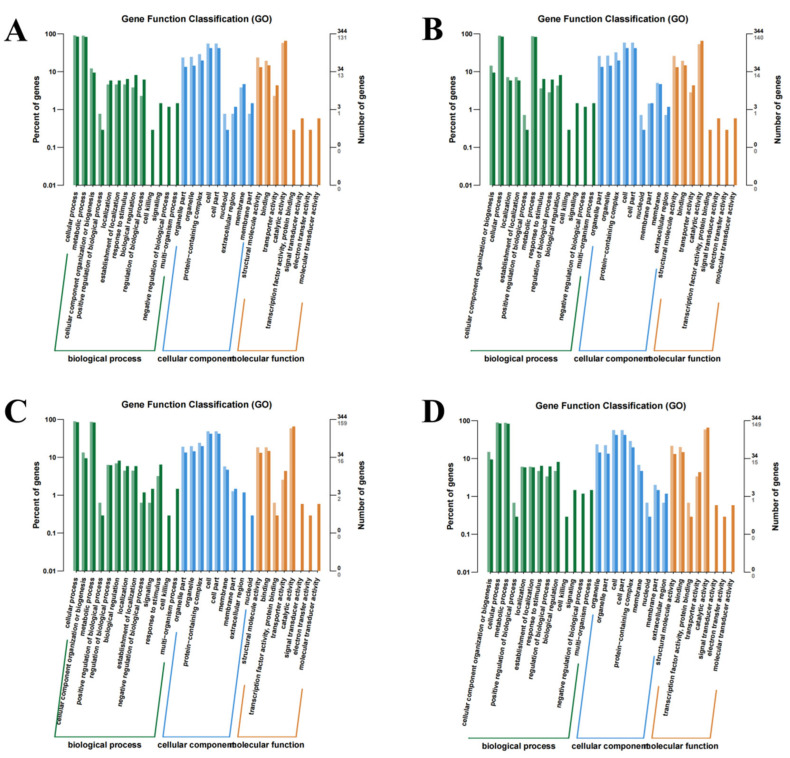
Column-like diagram of different gene GO comment classification. The horizontal axis represents various functional classifications, while the vertical axis displays the number of genes belonging to each classification (on the right) and the percentage of these genes in the total annotated genes (on the left). By using different colors, we can distinguish different classifications. The figures shows the functional annotation of DEGs between the (**A**) H2 and H1 groups; (**B**) L2 and L1 groups; (**C**) C2 and C1 groups; and (**D**) N2 and N1 groups.

**Figure 5 microorganisms-13-01173-f005:**
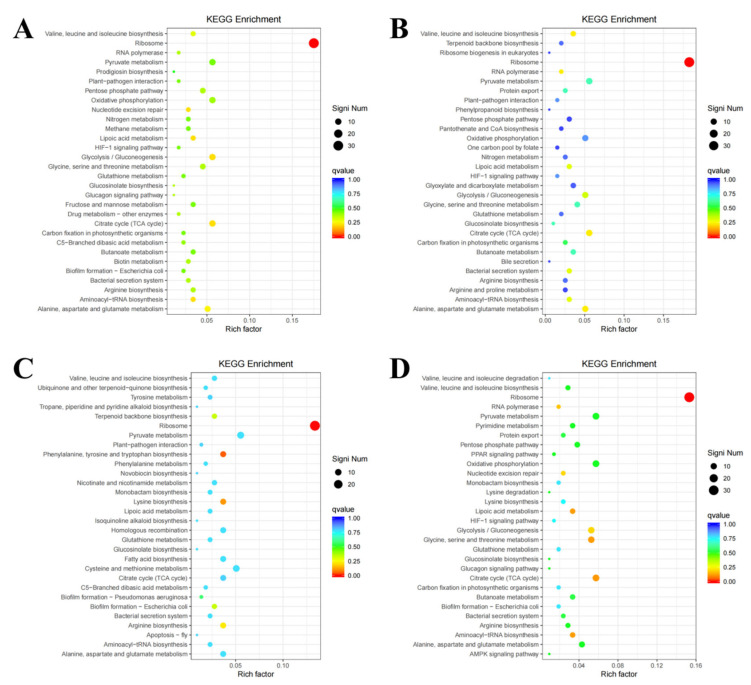
Significant enrichment function scatter map. The vertical axis displays feature annotation information, while the horizontal axis represents the rich factor value of the feature. The color of the dot represents the size of the Qvalue, where the closer the color is to red, the smaller the Qvalue. The number of DEGs included in each function is reflected by the dot size. The figure shows the scatter plots of significant enrichment functions between the (**A**) H2 and H1 groups; (**B**) L2 and L1 groups; (**C**) C2 and C1 groups; and (**D**) N2 and N1 groups.

**Figure 6 microorganisms-13-01173-f006:**
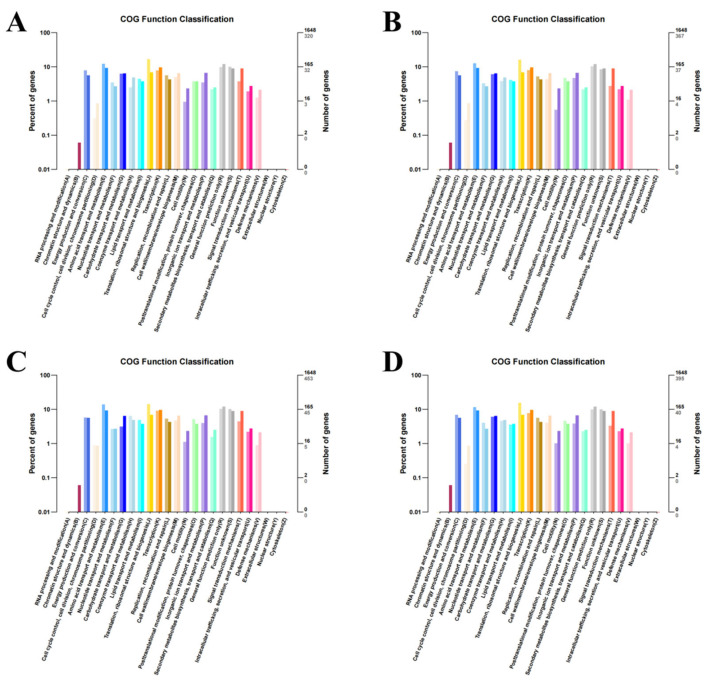
Bar chart of COG classification for DEGs. The horizontal axis represents COG functional categories, while the vertical axis displays the number of genes belonging to each classification (on the right) and the percentage of these genes in the total annotated genes (on the left). The figure illustrates the COG functional annotation of DEGs between the (**A**) H2 and H1 groups; (**B**) L2 and L1 groups; (**C**) C2 and C1 groups; and (**D**) and N1 groups.

**Figure 7 microorganisms-13-01173-f007:**
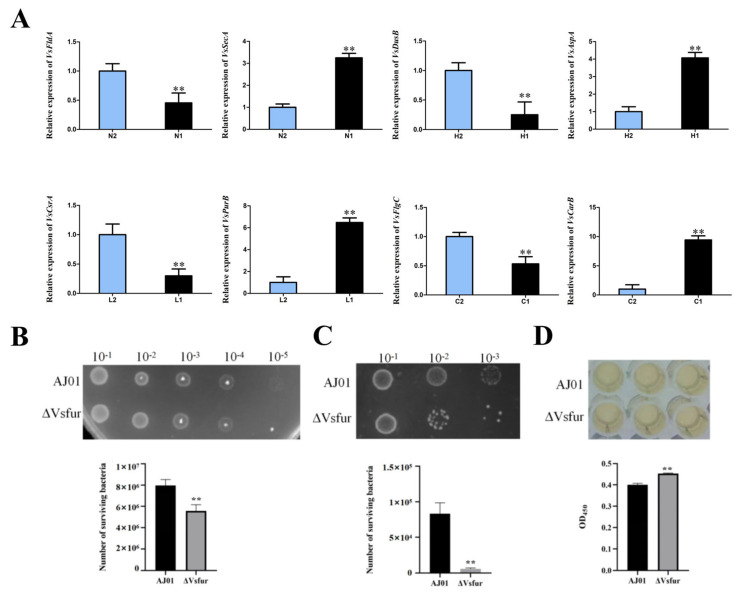
(**A**) Temporal expression analysis of *VsFldA*, *VsSecA, VsDusB, VsAspA, VsCrsA, VsPurB*, *VsFlgC,* and *VsCarB* in the H, L, C, and N groups. Values are presented as mean ± SD (*n* = 5). Asterisks indicate significant differences: ** *p* < 0.01. (**B**) Oxidative stress capacity of the two experimental strains AJ01 and Δ*Vsfur*. (**C**) Interbacterial competitiveness of the two experimental strains AJ01 and Δ*Vsfur*. (**D**) Cytotoxicity of the two experimental strains AJ01 and Δ*Vsfur*. The error line is the standard deviation (*n* = 3). The asterisks represent significant differences (** *p* < 0.01).

**Table 1 microorganisms-13-01173-t001:** Primers used in this study.

Primer	Sequences (5′–3′) ^a^
8F	AGAGTTTGATCCTGGCTCAG
1492R	GGTTACCTTGTTACGACTT
VsfurF	CTCGAGATGTCAGACAATAATCAAGCG (*Xho* I)
VsfurR	GGATCCAGTTACAATGCCAGCATC (*Bam*H I)
VsFldARTF	CTTGTTGCTATCTTTGGTTGTGG
VsFldARTR	ATTGGCTGTCATCGCCTTCA
VsSecARTF	AGGTTGGTTAGACGAAGACGACA
VsSecARTR	CGCGAACTTTAGAAAGGATGGT
VsDusBRTF	ATGGCTGGCGTAACGGATAG
VsDusBRTR	ACTGAACGAATGCCCGACTC
VsAspARTF	AACCGCAGCCAATCTACAAA
VsAspARTR	CCGTCATAGGCACAGCATCTT
VsCsrARTF	TGTTGGCGAAACACTGATGAT
VsCsrARTR	GGTGAACAGATACTTCTTTAGGTGC
VsPurBRTF	CCCAGAAGTTGAATGGCACC
VsPurBRTR	TTACGAAGAACCGTAGAGTCAGTAAG
VsFlgCRTF	TGGTTCTGCGATGAGTGCTG
VsFlgRTR	GCTAATGGGTGATCTGGGTTGTA
VsCarBRTF	ACTCGATCACAGTGGCTCCG
VsCarBRTR	CATAACTTCACCAACCGACTTCAT
933F	GCACAAGCGGTGGAGCATGTGG
16SRTR1	CGTGTGTAGCCCTGGTCGTA

^a^ Underlined nucleotides are restriction sites of the enzymes indicated in brackets at the end of the sequences.

**Table 2 microorganisms-13-01173-t002:** The list of data output quality.

Sample Name	Raw Reads	Clean Reads	Clean Bases	Error Rate (%)	Q20 (%)	Q30 (%)	GC Content (%)
H1	18,511,874	17,811,848	2.58 G	0.04	98.70%	96.69%	46.64%
H2	15,482,650	14,958,380	2.17 G	0.03	98.73%	96.79%	45.75%
L1	18,457,720	17,780,906	2.56 G	0.04	98.70%	96.71%	47.09%
L2	18,578,108	17,928,076	2.59 G	0.03	98.68%	96.67%	46.05%
N1	17,714,246	17,072,464	2.47 G	0.04	98.61%	96.41%	46.90%
N2	18,152,356	17,497,854	2.50 G	0.04	98.69%	96.71%	46.20%
C1	60,946,986	58,627,030	8.44 G	0.04	98.58%	96.08%	54.91%
C2	62,823,344	60,461,710	8.60 G	0.04	98.61%	96.14%	53.99%

## Data Availability

The original contributions presented in this study are included in the article. Further inquiries can be directed to the corresponding authors.

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
