# Peer review of "RNA-Seq Analysis Revealed the Virulence Regulatory Network Mediated by the Ferric Uptake Regulator (Fur) in Apostichopus japonicus Pathogenesis Induced by Vibrio splendidus"

_microorganisms, 2025, doi:10.3390/microorganisms13061173_

Round 1

Reviewer 1 Report

Comments and Suggestions for Authors

Dear authors, here are some comments and suggestions:
- First of all, the manuscript does not meet the journal's standards. It is necessary to comply with the standards.
- Look for keywords that are different from the terms in the title to increase the indexing potential of the manuscript.
- The text has too many acronyms, which make it difficult to read, and I recommend removing all acronyms created by the authors. Leave only those that are universally agreed upon.

- The text has several redundant and repetitive parts. It is a matter of style, but if the authors are concerned about how their target audience will evaluate the manuscript, it would be interesting to review the language and structure. In the introduction, we see several repetitions of topics in the paragraphs. The results begin by describing the objectives of the study again, as well as parts of the methodology. Again, to make the text more readable, appreciable and even legible so that it can be cited, this structural review would be necessary.

- The authors begin the results with the cited repetitions and then refer to Table 1 (Line 65). The authors should remove the text from lines 257-262 and start the text directly indicating the number of raw and reads, pointing out the main results presented in Table 1, and then make the call.
- Why are there two terms in Portuguese on lines 263-264? - The figures are excellent, but especially the ones written on the image are of inadequate resolution, but I believe the authors will send a high-resolution version for publication.
- Lines 358-361: Because the authors are mentioning the study's objectives, again in the discussion.

- Line 387: Mention examples.
- Line 389: splendides?
- The text has an adequate method, coherent interpretations and lasting conclusions. It has some structural problems, but it has a lot of potential.

Author Response

Comments 1: First of all, the manuscript does not meet the journal's standards. It is necessary to comply with the standards.

Response 1: We thank the reviewer’s valuable comments. We have revised the manuscript according to the journal's formatting standards and supplemented information such as the author contributions section.

Comments 2: Look for keywords that are different from the terms in the title to increase the indexing potential of the manuscript.

Response 2: We appreciate the reviewer's suggestive comments. The keywords have been modified as suggested. Or you can see that in the revised manuscript in the “Keywords” Line 35 and 36.

Comments 3 The text has too many acronyms, which make it difficult to read, and I recommend removing all acronyms created by the authors. Leave only those that are universally agreed upon.

Response 3: We agree with the reviewer’s suggestions and removed unnecessary acronyms, thereby enhancing the clarity and accessibility of the manuscript. Or you can see that in Line 70, 73, 328, 420, 452, 480Comments :483.

Comments 4: The text has several redundant and repetitive parts. It is a matter of style, but if the authors are concerned about how their target audience will evaluate the manuscript, it would be interesting to review the language and structure. In the introduction, we see several repetitions of topics in the paragraphs. The results begin by describing the objectives of the study again, as well as parts of the methodology. Again, to make the text more readable, appreciable and even legible so that it can be cited, this structural review would be necessary.

Response 4: Many thanks for your comments. We have carefully checked throughout the manuscript and removed any redundant and repetitive content. Or you can see that in “Introduction” Line 85 and “Results” Line 247Comments :248.

Comments 5: The authors begin the results with the cited repetitions and then refer to Table 1 (Line 65). The authors should remove the text from lines 257Comments :262 and start the text directly indicating the number of raw and reads, pointing out the main results presented in Table 1, and then make the call.

Response 5: We agree with the reviewer’s suggestions. As mentioned earlier, we have followed your suggestions and removed the redundant citations in Section 3.1 of the Results. Or you can see that in Line 247Comments :248.

Comments 6: Why are there two terms in Portuguese on lines 263-264? The figures are excellent, but especially the ones written on the image are of inadequate resolution, but I believe the authors will send a high-resolution version for publication.

Response 6: We appreciate the reviewer’s suggestion. We have uploaded the original high-resolution figures.

Comments 7: Lines 358-361: Because the authors are mentioning the study's objectives, again in the discussion.

Response 7: We are sorry for the errors. We have corrected the mistakes. Or you can see that in the revised manuscript in the “Discussion” Line 405-407.

Comments 8: Line 387: Mention examples.

Response 8: We are sorry for the missing information. Based on your suggestions, the detailed information has been added in the revised manuscript. Or you can see that in “Discussion” Line 429-433.

Comments 9: Line 389: splendides?

Response 9: We thank the reviewer for pointing our mistakes. We have carefully checked throughout the manuscript and replace “splendides” by “splendidus”. Or you can see that in the revised manuscript in the “Discussion” Line 434.

Comments 10: The text has an adequate method, coherent interpretations and lasting conclusions. It has some structural problems, but it has a lot of potential.

Response 10: We thank the reviewer for your comments and have modified the paper following your suggestion. We hope the revised manuscript meets your expectations.

Reviewer 2 Report

Comments and Suggestions for Authors

The manuscript presents a strong and detailed transcriptomic investigation of the virulence regulatory network mediated by the ferric uptake regulator (Fur) in Vibrio splendidus during infection of Apostichopus japonicus.

The study is highly relevant, as it provides new insights into how Fur controls iron-dependent and pathogenicity-related gene expression.

The simulation of host-pathogen interaction using sea cucumber coelomic fluid is scientifically sound and adds biological relevance to the findings.

The identification of key virulence pathways, such as flagellar assembly, bacterial secretion systems, and oxidative stress resistance, is well supported by both RNA-seq and qRT-PCR validation.

The phenotypic assays, including oxidative stress, bacterial competition, and cytotoxicity, greatly strengthen the conclusions by linking transcriptomic changes to functional outcomes.

Overall, the study advances understanding of Fur-mediated virulence regulation in marine pathogens and provides a strong basis for future therapeutic strategies targeting Vibrio splendidus.

Only minor suggestions: streamline a few repetitive parts in the discussion, and emphasize the novelty compared to previous studies on Fur in Vibrio species.

Comments on the Quality of English Language

The manuscript is generally well-written in clear and professional English.

The language is fluent, with appropriate scientific terminology and logical flow between sections.

Only minor polishing is recommended to improve clarity:

  • Shorten a few overly long sentences to make the text easier to read.
  • Eliminate minor repetition in some parts of the introduction and discussion.
  • Carefully check a few transitional phrases between paragraphs to improve the flow.

Author Response

Comments 1: Only minor suggestions: streamline a few repetitive parts in the discussion, and emphasize the novelty compared to previous studies on Fur in Vibrio species.

Response 1: We agree with the reviewer’s suggestions. We have streamlined redundant content in the discussion section and enhanced the description of this study's novel contributions. Or you can see that in the revised manuscript in the “Discussion” Line 400-407 and 429-436.

Comments 2: Comments on the Quality of English Language:

The manuscript is generally well-written in clear and professional English.

The language is fluent, with appropriate scientific terminology and logical flow between sections.

Only minor polishing is recommended to improve clarity:

Shorten a few overly long sentences to make the text easier to read.

Eliminate minor repetition in some parts of the introduction and discussion.

Carefully check a few transitional phrases between paragraphs to improve the flow.

Response 2: We thank the reviewer for their valuable suggestions and have revised the manuscript accordingly. The manuscript has been professionally edited by MDPI's language service (ID: english-edited-93775).

Reviewer 3 Report

Comments and Suggestions for Authors

This manuscript presents a comprehensive RNA-seq-based analysis of the ferric uptake regulator (Fur) and its role in regulating virulence in Vibrio splendidus during infection of Apostichopus japonicus. The study is timely and relevant for understanding host-pathogen interactions in marine aquaculture. The combination of transcriptomics, gene knockdown, qRT-PCR validation, and functional assays provides robust support for the conclusions.

The study addresses an important aquaculture disease and contributes significantly to our understanding of the molecular pathogenesis of V. splendidus. The experimental design is well-structured, with clear comparisons across iron availability conditions and between wild-type and ΔVsfur strains. The integration of transcriptomic data with phenotypic assays strengthens the biological interpretation, and the conclusions are well-supported by the data.

Several sections would benefit from careful English language editing to improve clarity, reduce redundancy, and polish sentence structure. This includes both the main text and the figure captions. Please ensure that all figures, especially heatmaps and pathway enrichment plots, are clearly labeled, legible, and described in sufficient detail in the legends. Axis labels and gene identifiers should be easy to read.

While statistical methods are mentioned, more specific details should be provided, such as the methods used for multiple testing correction and clear thresholds for differential gene expression. The discussion section could be streamlined to better highlight the main findings and their broader significance. A more structured discussion of key results, biological relevance, and implications would improve readability.

Adding a brief note on how this research could inform future prevention or treatment strategies for skin ulcer syndrome in aquaculture would strengthen the practical relevance of the work.

Comments on the Quality of English Language

The manuscript is generally understandable and scientifically accurate, but the quality of English can be improved. There are frequent issues with sentence structure, word choice, and clarity that occasionally hinder readability. A thorough language revision by a native English speaker or professional editing service is recommended to enhance the clarity and flow of the text.

Author Response

Comments 1: Several sections would benefit from careful English language editing to improve clarity, reduce redundancy, and polish sentence structure. This includes both the main text and the figure captions. Please ensure that all figures, especially heatmaps and pathway enrichment plots, are clearly labeled, legible, and described in sufficient detail in the legends. Axis labels and gene identifiers should be easy to read.

Response 1: We thank the reviewer for your constructive comments. In the revised manuscript, we have performed language editing on the main text and figure captions, and uploaded the original high-resolution figures.

Comments 2:While statistical methods are mentioned, more specific details should be provided, such as the methods used for multiple testing correction and clear thresholds for differential gene expression. The discussion section could be streamlined to better highlight the main findings and their broader significance. A more structured discussion of key results, biological relevance, and implications would improve readability.

Response 2: Many thanks for your comments. We have supplemented and annotated the statistical analysis methods in the revised manuscript. Or you can see that in the “Materials and Methods” Line 183-184 and 237-244. In accordance with your recommendations, we have revised the manuscript and streamlined the discussion section by removing redundant content. Or you can see that in the revised manuscript in the “Discussion” Line 400-407, 409-411 and 437-439.

Comments 3:Adding a brief note on how this research could inform future prevention or treatment strategies for skin ulcer syndrome in aquaculture would strengthen the practical relevance of the work.

Response 3: We agree with the reviewer's suggestion and have inserted the prevention and control strategies at the end of the conclusions in our revised manuscript.

Comments 4: Comments on the Quality of English Language:

The manuscript is generally understandable and scientifically accurate, but the quality of English can be improved. There are frequent issues with sentence structure, word choice, and clarity that occasionally hinder readability. A thorough language revision by a native English speaker or professional editing service is recommended to enhance the clarity and flow of the text.

Response 4: We appreciate the reviewer’s comment and improve the language by MDPI's language service (ID: english-edited-93775).

Round 2

Reviewer 1 Report

Comments and Suggestions for Authors

The changes took into account my suggestions.